# Effectiveness and Components of Web-Based Interventions on Weight Changes in Adults Who Were Overweight and Obese: A Systematic Review with Meta-Analyses

**DOI:** 10.3390/nu15010179

**Published:** 2022-12-30

**Authors:** Yutong Shi, Kyohsuke Wakaba, Kosuke Kiyohara, Fumi Hayashi, Kazuyo Tsushita, Yoshio Nakata

**Affiliations:** 1Graduate School of Comprehensive Human Sciences, University of Tsukuba, Tsukuba 305-8574, Japan; 2Faculty of Human Life, Jumonji University, Niiza 352-8510, Japan; 3Department of Food Science, Faculty of Home Economics, Otsuma Women’s University, Chiyoda-ku 102-8357, Japan; 4Faculty of Nutrition, Kagawa Nutrition University, Sakado 350-0288, Japan; 5Faculty of Health and Sport Sciences, University of Tsukuba, Tsukuba 305-8574, Japan

**Keywords:** internet, web, obesity, weight loss

## Abstract

Overweight and obesity have reached epidemic proportions worldwide. The COVID-19 pandemic resulted in an increased need for remote implementation of weight-loss interventions; therefore, the effectiveness of web-based interventions needed to be assessed. This study aimed to examine the effectiveness of web-based interventions and their potency in facilitating weight changes in adults who were overweight or obese. We searched PubMed and Ichu-shi Web from the first year of inclusion in each database until the search date (30 September 2020). Among 1466 articles retrieved from the two databases and manual search, 97 were selected to undergo qualitative analysis and 51 articles were subjected to quantitative analysis. Qualitative analysis of 97 articles demonstrated that articles showing significant effectiveness mostly used the following components: social support, self-monitoring for behavior, self-monitoring for the outcome (weight), behavioral goal setting, information about health consequences, and outcome goal setting. Quantitative analysis of 51 articles showed a significant effectiveness of web-based intervention (standardized mean difference, −0.57; 95% confidence interval, −0.75 to −0.40). This study demonstrated the effectiveness of web-based interventions on weight change in adults with overweight and obesity. Subgroup meta-analyses identified personalized information provision and expert advice to be remarkably effective components.

## 1. Introduction

Concern regarding obesity and overweight is increasing worldwide in several fields. These conditions pose a substantial threat for developing serious chronic diseases, including type 2 diabetes, cardiovascular disease, hypertension, sleep apnea, osteoarthritis, and certain cancers [1]. Thirty-nine percent of adults worldwide are overweight (25 kg/m^2^ ≤ body mass index [BMI] < 30 kg/m^2^) with 13% being obese (BMI ≥ 30 kg/m^2^) [1]. An imbalance in energy causes obesity; hence, behavioral changes can improve energy balance. The guideline for the management of overweight and obesity in adults by the American Heart Association, American College of Cardiology, and the Obesity Society shows that conventional weight-loss interventions through lifestyle modification are effective; however, examination of the relative effectiveness and related charges of delivering interventions on site (face-to-face), remotely or by a mixture of approaches, is required [2]. Providing lifestyle interventions through the Internet, mobile phones, text messaging, telephones, or a combination of these is a potentially cost-effective and compelling approach [2]. As the COVID-19 pandemic has raised awareness about the necessity of implementing weight-loss interventions remotely, the effectiveness of web-based interventions should be assessed.

Information and communication technology is developing rapidly. As of 2021, about 4.88 billion (61.8%) of the 7.9 billion people worldwide used the Internet, and the number of Internet users increased by about 220 million (4.8%) from 2020 to 2021 [3]. Consequently, web-based interventions have become widely utilized to promote good health. Nevertheless, the impact of these interventions on health outcomes, evaluated using valid methodology, is not always reported. The use of fitness-related technologies, such as wearable devices and smartphone apps, has been suggested to sustainably increase physical activity, and they play an essential role in public health [4]. Such technologies can remove many of the barriers to large-scale interventions [4] and tailor interventions to the needs of individuals and specific populations [5]. A systematic review of web-based weight-loss interventions recently found them to be effective in comparison to a non-technology active or inactive (wait-list) control group [6]. Additionally, the methodologies of the studies used in this review are diverse and include multiple components: computers, smartphone apps, and online chats.

Therefore, which components are most effective among web-based interventions remains unclear. This systematic review with meta-analyses aimed to summarize the effectiveness of web-based weight-loss interventions and examine the effective components qualitatively and quantitatively.

## 2. Materials and Methods

This systematic review followed the principles of the Preferred Reporting Items for Systematic Reviews and Meta-Analyses Statement (PRISMA) [7] and Cochrane Collaboration guidance [8]. Prior to initiating the review, we searched the International Prospective Register of Systematic Reviews (PROSPERO) and found no existing reviews for this issue. The study protocol was registered with PROSPERO on 6 January 2021 (CRD42021218570).

### 2.1. Search Methods

To search for articles, an optimal search formula was created by combining keywords and thesauri (MeSH in PubMed) according to the clinical question set in advance. The search was conducted using PubMed and Ichu-shi Web, which accumulates medical literature published in Japan and contains more than 13 million articles [9]. The search period was from the first year of inclusion of each database to the search date (30 September 2020). The following search terms were identified for each relevant category: population, intervention, outcome, and publication type. The search terms and formulas for PubMed and Ichu-shi Web are listed in Appendix A, respectively.

### 2.2. Criteria for Exclusion

The criteria for exclusion from the study were as follows: (1) the language used was not English or Japanese; (2) not an original paper; (3) non-human participants; (4) all participants were under the age of 18 years; (5) all participants had a BMI < 25 kg/m^2^; (6) targeted pregnant women; (7) not a randomized controlled trial; (8) not a web-based intervention; (9) weight change was not reported; (10) included only data on weight maintenance; and (11) other conditions based on the discretion of the investigators.

### 2.3. Selection Process and Data Extraction

Two researchers (Y.S. and K.W.) independently screened the records obtained from the literature search to determine their eligibility. Titles and abstracts were screened first, and the full texts were extracted and checked to determine the eligibility of the journal articles. Any inconsistency between the two reviewers was resolved by discussing it with a third reviewer (Y.N.). We extracted the following information from the accepted papers: (1) author(s), year, and country of publication; (2) intervention medium (computer or smartphone); (3) percentage of male participants; (4) completion rate; (5) mean BMI; (6) mean age; (7) intervention period; and (8) significance of the results. In addition, for the articles included in the meta-analysis, we extracted the sample size, primary outcome (weight change), and specific methods of interventions for each group of studies.

### 2.4. Data Synthesis

#### 2.4.1. Quality of Evidence

The eligible studies were evaluated for the risk of bias using a revised Cochrane risk of bias tool [8]. This consisted of evaluating the risk of selection (randomization and allocation), performance (blinding of participants and personnel), detection (blinding of outcome assessment), attrition (incomplete outcome data), reporting (selective reporting), and other biases of individual studies. There were three levels for each item (high, low, and unknown) and two reviewers (Y.S. and K.W.) independently evaluated each paper. Any disagreement was determined by discussion with a third reviewer (Y.N.).

#### 2.4.2. Qualitative Analysis

In the qualitative analysis, the components of the intervention content of the articles were listed and coded using behavior change technique (BCT) taxonomy (v1) [10], which is a taxonomy methodology comprising 93 individual BCTs grouped in 16 hierarchical clusters (e.g., goals and planning, feedback and monitoring, social support, and shaping knowledge). It is a reliable method that could specify content in terms of BCTs: the smallest components of behavior change interventions which, on their own and in favorable circumstances, can facilitate change [10].

One researcher (Y.S.) was responsible for coding, collecting, summarizing, and analyzing the frequency distributions of BCTs, and another researcher (K.W.) confirmed the content. Then, we compared the frequency of BCTs among articles that were and were not employed in the meta-analysis and those that had or did not have a significant effect compared to the control group to determine which BCTs were likely to display a higher efficacy.

#### 2.4.3. Quantitative Analysis

The results of meta-analyses were presented as forest plots and funnel plots. Review Manager 5.4 (RevMan, The Cochrane Collaboration, Oxford, UK) was used for the analysis. We used a random-effects model to calculate the standardized mean difference (SMD) and 95% confidence intervals (CIs) for weight changes (data given in pounds were converted to kg). *I*^2^ determined the degree of heterogeneity among the study results. Therefore, articles that did not provide data to allow calculation of SMD, such as standard deviation (SD), standard error (SE), or 95% CIs, were not included in the meta-analysis. The calculated SMD was interpreted according to Cohen’s effect size [11].

#### 2.4.4. Subgroup Analysis

In addition to the meta-analysis that examined the effectiveness of web-based intervention on weight changes, subgroup analyses were performed to explore the effective components. Other subgroup analyses were also conducted to evaluate the intervention medium, publication year, and intervention period. To analyze comparable constructs, a meta-analysis was performed when data could be extracted from two or more articles.

## 3. Results

### 3.1. Description of Included Studies

The search strategy resulted in 1466 articles being retrieved from the two databases (1305 from PubMed and 160 from Ichu-shi), with one manually searched article. After screening, 97 studies were included (Figure 1), published from 2001 to 2020. Of these, 18, 41, and 38 papers were published in 2001–2010, 2011–2015, and 2016–2020, respectively. The United States was the most surveyed country (*n* = 57), followed by Australia (*n* = 10), the United Kingdom (*n* = 8), Japan (*n* = 6), the Netherlands (*n* = 5), Spain (*n* = 3), and Iran, Canada, Turkey, Finland, France, Poland, South Korea, and Hong Kong (*n* = 1 each). Seventy-two and 25 studies used computers and smartphone applications as intervention media, respectively. The mean age of the participants in each study was 20.5–69.0 years. The mean BMI was 26.1–38.1 kg/m^2^, except for five studies wherein it was not noted. The proportion of male participants was 0–100%, except for eight studies wherein the participants’ sex was not stated. The intervention duration, described in all articles, was 2–24 months. Completion rates ranged from 16.4% to 100%, except in five studies in which the rate of completion was not stated. The specific features of the retrieved studies are presented in Appendix A.

### 3.2. Quality of the Evidence

Regarding risk of bias for eligible studies, over half the studies reported randomization with sequence generation. Two studies did not conceal the allocation. As expected, blinding participants in this type of intervention was not feasible, with six cases of assessor blindness not being reported in the article. Most studies had a low risk of bias in attrition and reporting. The main sources of high risk for other biases included block randomization, non-blind randomization, and baseline imbalance.

In the risk of bias assessment, 1 article [12] was judged to be high risk (−2), 53 articles were medium risk/suspected risk (−1), and 43 articles were low risk (0) (Appendix A). Among the endpoints, the highest number of studies regarded as high risk (−2) or medium risk/suspected risk (−1) were those with failure to blind participants and personnel (84.5%) and an inability to describe allocation concealment (70.1%).

### 3.3. Qualitative Analysis

For the qualitative analysis, we categorized the components of the web-based intervention for the 97 articles (Appendix A); each article had 2–10 (5.2 ± 1.8) intervention components. Each article was included in one of three groups: “significantly different,” “not significantly different,” or “not included in meta-analysis” in comparison to the control group. Figure 2 displays the number of articles that included each intervention component in order of descending total number. The most frequently used components were self-monitoring of behavior (*n* = 76), social support [unspecified] (*n* = 66), self-monitoring of outcome of behavior (*n* = 63), goal setting [behavior] (*n* = 53), and information about health consequences (*n* = 53). However, the articles that showed a significant effectiveness contained more social support [unspecified] (24/66), self-monitoring of behavior (24/76), self-monitoring of outcome of behavior (21/63), goal setting [behavior] (17/53), information about health consequences (16/53), and goal setting [outcome] (13/31).

### 3.4. Quantitative Analysis

Among the 97 articles, 51 for which data could be extracted were subjected to meta-analysis. The characteristics of the papers retrieved in the meta-analysis are presented in Appendix A [13,14,15,16,17,18,19,20,21,22,23,24,25,26,27,28,29,30,31,32,33,34,35,36,37,38,39,40,41,42,43,44,45,46,47,48,49,50,51,52,53,54,55,56,57,58,59,60,61,62] according to the year of publication in ascending order. Among the 51 articles from which data were extracted, a meta-analysis was performed using 44 articles that could compare the effects of web-based interventions with offline control groups. Figure 3 shows the forest plot; the left-hand column lists the authors’ names from the studies and year of publication in ascending order, and the right-hand column is a plot of the measure of effect (SMD) for each of these studies, incorporating CIs represented by horizontal lines. The meta-analyzed measure of effect is plotted as a diamond, the lateral points of which indicate CIs for this estimate. The SMD was −0.57, the 95% CI was −0.75 to −0.40, and *I^2^* was 91%. Figure 4 illustrates the funnel plot to graphically assess any potential publication bias. The studies were plotted with the estimated effect on the horizontal axis and the SE of the estimated effect on the vertical axis. On the horizontal axis, the further to the left, the more effective the intervention group is, and the further to the right, the more effective the control group is. On the vertical axis, the larger sample size, the higher on the graph the study is plotted, and the smaller sample size, the lower the study is plotted. In the absence of a publication bias, the plot should look symmetric.

### 3.5. Subgroup Analyses

#### 3.5.1. Intervention Medium (Computer or Smartphone)

Of the 44 articles that could be analyzed quantitatively, 29 articles provided web-based interventions through computers and 15 through smartphone apps. The web-based interventions through both computers (SMD, −0.49; 95% CI, −0.70 to −0.28; *I*^2^ = 93%) and smartphone apps (SMD, −0.75; 95% CI, −1.06 to −0.44; *I^2^* = 85%) were significantly effective, as illustrated in Appendix A, respectively.

#### 3.5.2. Publication Year

The publication years of the articles included in this study were from 2001 to 2020. A meta-analysis performed with papers published from 2001 to 2010 indicated that web-based interventions had no effect (SMD, −0.28; 95% CI, −0.94 to 0.38; *I*^2^ = 97%). The web-based interventions published from 2011 to 2015 (SMD, −0.72; 95% CI, −0.95 to −0.50; *I*^2^ = 87%) and from 2016 to 2020 (SMD, −0.50; 95% CI, −0.74 to −0.25; *I*^2^ = 87%) were significantly effective, as illustrated in Supplementary Appendix A.

#### 3.5.3. Intervention Period

Remarkable impact was observed for web-based intervention lasting < 3 months (SMD, −0.70; 95% CI, −0.98 to −0.43; *I*^2^ = 23%), 3–6 months (SMD, −0.66; 95% CI, −0.88 to −0.44; *I*^2^ = 88%), and > 6 months (SMD, −0.38; 95% CI, −0.71 to −0.05; *I*^2^ = 95%) as illustrated in Supplementary Appendix A.

#### 3.5.4. Intervention Component

The effectiveness of three components of web-based intervention for which data could be extracted from more than one article was analyzed: personalized information, e-counseling, and online chats. The results of the meta-analysis indicated the effectiveness of personalized information (SMD, −0.39; 95% CI, −0.73 to −0.05; *I*^2^ = 86%) and e-counseling (SMD, −0.42; 95% CI, −0.75 to −0.08; *I*^2^ = 64%); however, potency of online chats was not observed (SMD, 0.18; 95% CI, −0.09 to 0.45; *I*^2^ = 68%), as illustrated in Figure 5, Figure 6 and Figure 7.

## 4. Discussion

This study is a systematic review and meta-analysis intended to establish the effectiveness of web-based interventions for weight change in people who were overweight or obese and examine the effective components qualitatively and quantitatively. The results showed the effectiveness of web-based interventions and two effective components: personalized information provision and e-counseling.

In the qualitative analysis, we summarized the 93 components of the web-based intervention using BCT taxonomy (v1) [10]. In the articles selected for this study, 21 components were applied. The present qualitative analyses showed that the significant interventions included social support, self-monitoring for behavior change, self-monitoring for the outcome (weight), behavioral goal setting, information provision, and outcome goal setting. Another similar qualitative review [63] that examined the effective components of web-based interventions showed that self-monitoring, feedback and communication, social support, use of structured programs, and use of individualized programs were factors promoting weight loss, which corresponds to our qualitative analysis findings.

Our quantitative analysis showed that web-based interventions were significantly effective (SMD, −0.57; 95% CI, −0.75 to −0.40). Cohen [11] interpreted 0.2 as a small effect, 0.5 as a moderate effect, and 0.8 as a large effect. Thereby, the effect of web-based intervention presented in this study can be interpreted as moderate. However, as shown in Figure 4, the funnel plot was not symmetrical, suggesting the presence of publication bias. That is, studies with smaller sample sizes that did not show significant effectiveness remained unpublished. Therefore, the effect size of web-based interventions in this study may have been overestimated.

The papers employed in this study were published through the years 2001 to 2020. As per Our World in Data [64], in 1990, few computers worldwide were connected to the Internet, and only 0.5% of the world’s population was estimated to have access to the Internet. Gradually, the number of Internet users began to increase, and by 2000, about half of the US population had access to the Internet. However, the Internet still had not made a significant impact in other parts of the world. Over the next 15 years, countries outside the US gained access to the Internet. In this study, about 90% of the countries surveyed in the papers were composed of the US and other Western countries, suggesting a possible selection bias.

In addition, subgroup analysis showed that the publication year affected the effectiveness of web-based intervention. Meta-analysis of papers published from 2001 to 2010 had no effect; however, the results of the papers published from 2011 to 2015 and 2016 to 2020 were significant. This difference may be due to the differing intervention media. Of the 97 papers included in this study, 72 used a computer as the intervention medium, while 25 utilized a smartphone application. Appendix A shows the specific features of the retrieved studies according to the year of publication in ascending order. Most older studies used computers as the intervention medium, whereas from 2013, some studies reported using smartphones. This transition may have increased the effectiveness of web-based interventions. The smartphone technology has enhanced computational functionality, which is ubiquitous and an acceptable technology platform. Computer-based interventions place a greater burden of self-monitoring on participants, which could be reduced by utilizing smartphones, owing to their notifying functions. The number of users of fitness apps has been increasing in recent years, with an estimated 60% increase in users between 2012 and 2017 [65]. This suggests that the number of papers using smartphone applications will proportionally increase in the future.

Additionally, the subgroup analysis for intervention duration showed that the web-based interventions were significantly effective in studies with intervention durations of less than 3 months, between 3 and 6 months, and more than 6 months, with SMDs of −0.70, −0.66, and −0.38, respectively. Although this does not corroborate the previous study [6] showing the effectiveness of short-term (<6 months) web-based interventions for weight loss in contrast with the lack of effectiveness of long-term interventions (≥6 months), this review indicates a trend toward decreasing effect with longer intervention durations. This suggests that the long-term use of web-based interventions should be considered when planning such interventions.

Furthermore, we conducted subgroup analyses to determine the effectiveness of the three components of the web-based intervention. While personalized information and e-counseling were significantly effective, online chats were ineffective. Therefore, the provision of personalized information tailored to the participant’s situation is likely to result in a weight-loss effect. In addition, web-based counseling from an expert during the intervention period may have increased the effectiveness of the intervention. A previous systematic review and meta-analysis [66] exhibits the effectiveness of behavioral therapy and counseling as components of web-based interventions, including individualized information and professional advice. However, only three articles from the same research group were included in the meta-analysis. The present study could have presented more substantial evidence, as five articles were adopted for personalized information and four were adopted for e-counseling. Chatting online did not appear to be effective. Since the online chats employed in the meta-analysis mostly took the form of interactions with other participants [17,20,50] or with a clinical psychologist [47], the lack of counseling and feedback from an expert may attenuate its effectiveness.

The strength of this study was that we included recently published papers that examined the effectiveness of web-based interventions, particularly considering the increasing number of papers on weight loss through web-based interventions every year. Additionally, the effective components of the web-based intervention were analyzed qualitatively and quantitatively utilizing BCT taxonomy (v1) [10]. In the future, the results of these analyses will allow us to test the effectiveness of new programs that combine effective components.

This systematic review had three limitations. The first limitation was the selection of articles. Only PubMed and Ichu-shi Web databases were used in this study. Additionally, the target languages were limited to English and Japanese. Therefore, we may not have extracted papers included in other databases or reported in other languages. In addition, “BMI” was not included in the search terms because this review focused on the effectiveness of web-based interventions on weight change. However, most relevant papers reported weight changes and BMI. We believe this may have prevented us from including all the papers in the analysis. Second, in the subgroup meta-analyses of this study, we could not examine all potential factors that may have influenced the effectiveness of web-based interventions. We conducted subgroup analyses focusing on the intervention medium, year of publication, intervention duration, and some intervention components; however, a previous study has mentioned that the frequency and duration of use of web-based programs may also affect the results [67]. In addition, some of the papers included in this study displayed the number of logins, self-monitoring sessions, and chat room participation as potential factors affecting the program effectiveness; however, subgroup analyses for these were not possible because of the shortage of related papers. Third, the present meta-analysis had a high degree of heterogeneity. The methods and duration of the interventions varied among the articles. Although a control group with no interventions would be ideal, most studies implemented minimal interventions, considering ethical issues. In addition, intervention programs included several intervention components, as shown in our qualitative analysis. Therefore, high heterogeneity may be inevitable in this issue. Finally, since this study excluded studies aimed at weight-loss maintenance, we could not explore the effects of web-based interventions and the effects of intervention components during the follow-up period. This is an aspect that needs to be considered in future studies.

## 5. Conclusions

This systematic review and meta-analysis demonstrated the impact of web-based interventions on weight change in adults with overweight and obesity. Subgroup meta-analyses identified the provision of individualized information and counseling from an expert as effective components. Further research is required to develop a new program that combines effective components and examines its effectiveness. In addition, the method used to disseminate and implement web-based interventions to the appropriate target population is a concern that should be addressed in the future.

## Figures and Tables

**Figure 1 nutrients-15-00179-f001:**
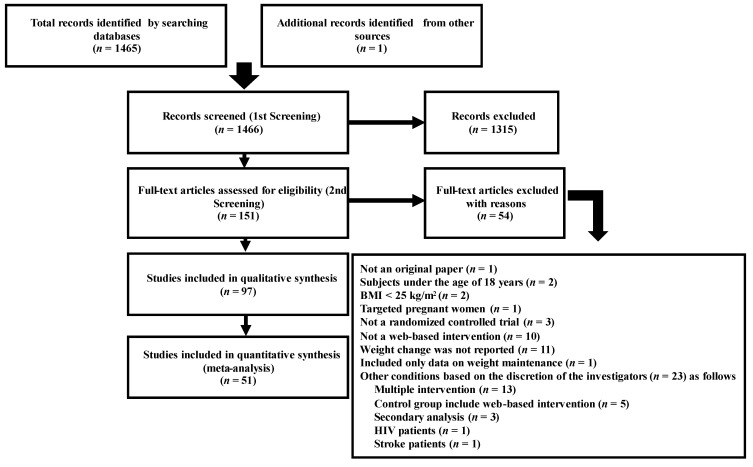
A flowchart illustrating the Preferred Reporting Items for Systematic Reviews and Meta-Analyses.

**Figure 2 nutrients-15-00179-f002:**
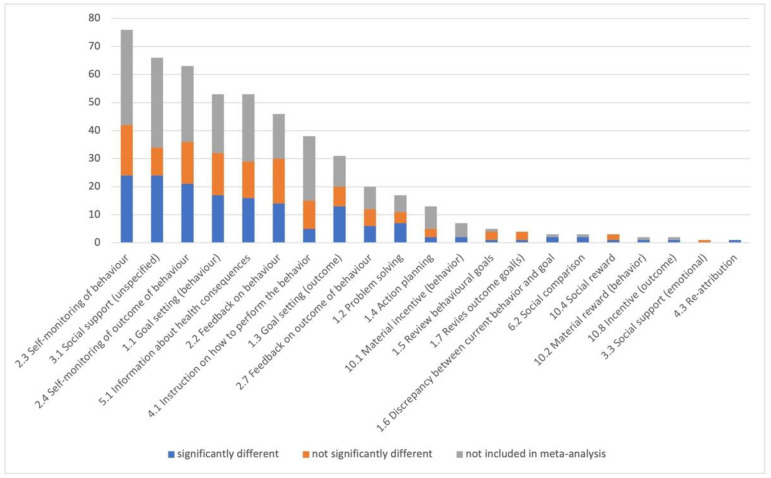
Number of articles including each intervention component.

**Figure 3 nutrients-15-00179-f003:**
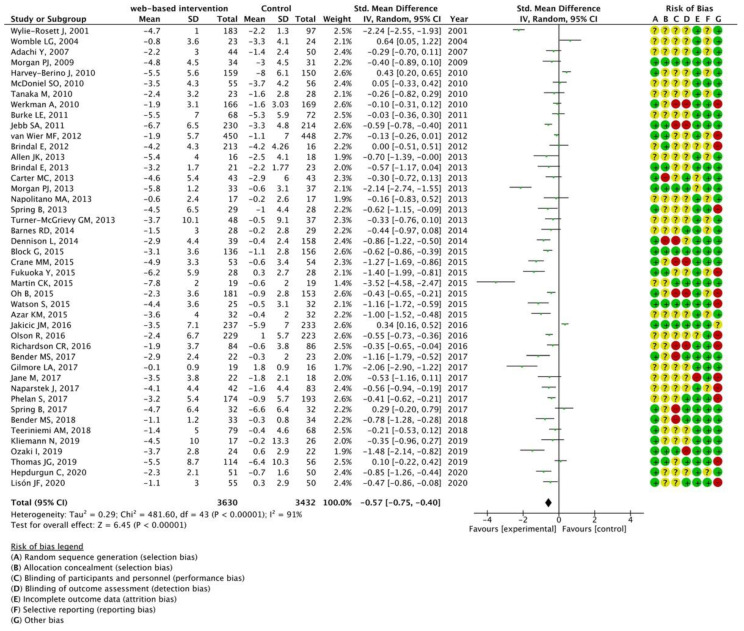
Meta-analysis for mean weight change comparing web-based interventions with offline control groups using 44 articles [12,13,15,17,20,21,22,23,24,25,26,27,28,29,30,32,33,34,35,36,37,38,39,40,41,42,43,44,45,46,47,49,50,51,52,53,54,55,56,58,59,60,61,62]. The forest plot shows standardized mean differences (SMD) for each article with 95% confidence intervals (CI). The diamond at the bottom of the graph means the meta-analyzed measure of effect and the lateral points of which indicate CIs for this estimate. A positive value reflects web-based interventions are more effective for weight loss than offline interventions. The area of each green square is proportional to the study's weight in the meta-analysis.

**Figure 4 nutrients-15-00179-f004:**
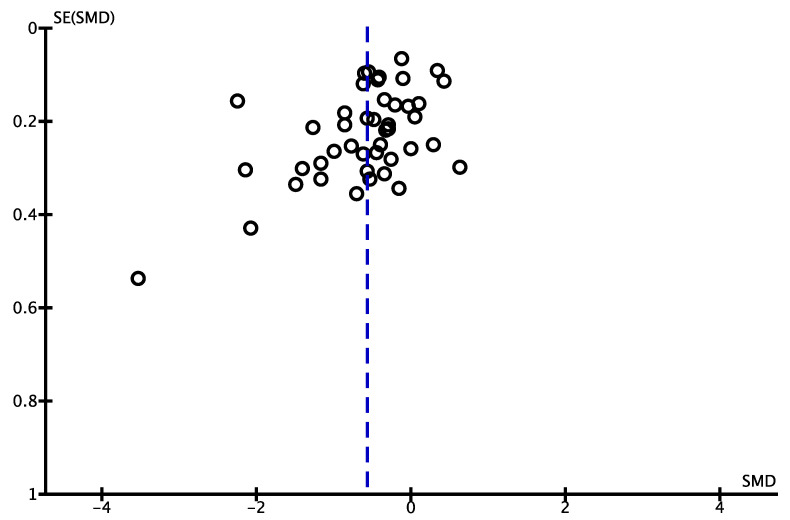
Funnel plot for mean weight change comparing web-based interventions with offline control groups (*n* = 44). Each circle represents a study, and the vertical dashed line represents the pooled effect size. In the absence of a publication bias, the plot should look symmetric. SMD, standardized mean difference; SE, standard error.

**Figure 5 nutrients-15-00179-f005:**
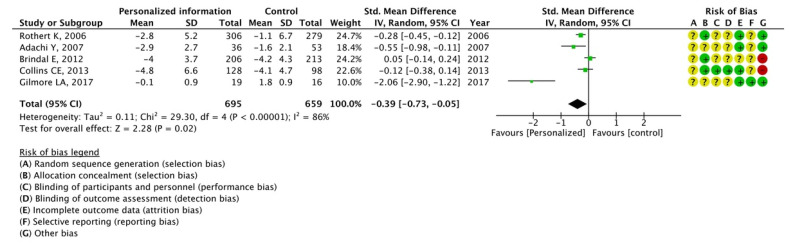
Meta-analysis for mean weight change with or without personalized information using 5 articles [16,17,26,31,50]. The forest plot shows standardized mean differences (SMD) for each article with 95% confidence intervals (CI). The diamond at the bottom of the graph means the meta-analyzed measure of effect and the lateral points of which indicate CIs for this estimate. A positive value reflects web-based interventions with personalized information are more effective for weight loss than web-based interventions without personalized information. The area of each green square is proportional to the study's weight in the meta-analysis.

**Figure 6 nutrients-15-00179-f006:**
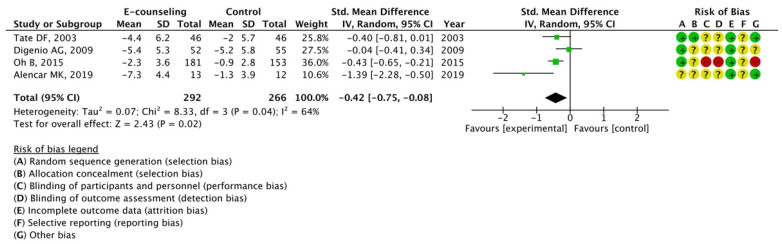
Meta-analysis for mean weight change with or without e-counseling using 4 articles [14,19,43,57]. The forest plot shows standardized mean differences (SMD) for each article with 95% confidence intervals (CI). The diamond at the bottom of the graph means the meta-analyzed measure of effect and the lateral points of which indicate CIs for this estimate. A positive value reflects web-based interventions with e-counseling are more effective for weight loss than web-based interventions without e-counseling. The area of each green square is proportional to the study's weight in the meta-analysis.

**Figure 7 nutrients-15-00179-f007:**
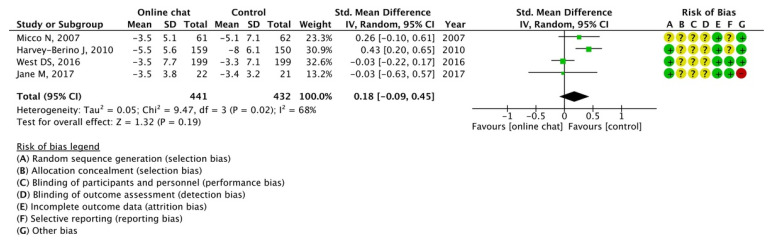
Meta-analysis for mean weight change with or without chatting online using 4 articles [18,21,48,51]. The forest plot shows standardized mean differences (SMD) for each article with 95% confidence intervals (CI). The diamond at the bottom of the graph means the meta-analyzed measure of effect and the lateral points of which indicate CIs for this estimate. A negative value reflects web-based interventions with online chat aren’t more effective for weight loss than web-based interventions without online. The area of each green square is proportional to the study's weight in the meta-analysis.

## Data Availability

Not applicable.

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
