# Peer review of "Effectiveness and Components of Web-Based Interventions on Weight Changes in Adults Who Were Overweight and Obese: A Systematic Review with Meta-Analyses"

_nutrients, 2022, doi:10.3390/nu15010179_

Round 1
Reviewer 1 Report
It's a systemic review with meta analyses, but it's too descriptive, so I'd like to have something presented. In subgroup analyses, authos descripted that papers published from 2001 to 2010 had no effect, and paper published from 2011 to 2015 and 2016 to 2020 had significant effect. I found it as very interesting point.
I wonder if there is any difference in the intervention component by year. It may have been difficult to analyze the frequency and duration of program use, but if there was no difference in components by year, authors can descript that simply creating a program would not help and steady usage of program is needed. Author can descript importace of enables steady access to program is important which can be solved by smart wath of SNS. I think this part could be added.
Reviewer 2 Report
The objective of the research is very pertinent given the importance that new technologies are acquiring in health interventions and specifically in weight loss programs, as has been revealed by the COVID19 pandemic that forced social distancing to large sectors of the population. population for many months.
METHODOLOGY
This is a systematic review conducted by two researchers. The methodology is adequate. The keywords used and the inclusion and exclusion criteria of the selected articles are well explained. It is a pity that BMI was not considered as an inclusion criterion
Validated and highly appropriate techniques have been used for the qualitative and quantitative analysis of the results.
RESULTS
The appendices document is not available on the platform (or I have not seen it). This makes it difficult to assess the quality of the evidence.
The graphics are very well designed and provide a lot of information. It would be interesting to explain that information in more detail in the text.
DISCUSSION
It is highly appreciated that the authors explain the limitations of their work, such as the absence of BMI as an inclusion criterion, this would also have allowed knowing the scope of the weight loss, that is, if these interventions serve to change the category of overweight or obesity at normal weight. Also the importance of the presence of a professional in the follow-up.
Round 2
Reviewer 1 Report
well done about revisioning manuscript.